# Promoting mechanism of serum amyloid a family expression in mouse intestinal epithelial cells

Masaki Wakai[1], Ryohei Hayashi[2]*, Yoshitaka Ueno[2], Kana Onishi[1], Takeshi Takasago[1], Takuro Uchida[3], Hidehiko Takigawa[2], Ryo Yuge[2], Yuji Urabe[1], Shiro Oka[1], Yasuhiko Kitadai[4], Shinji Tanaka[2]

1 Department of Gastroenterology and Metabolism, Hiroshima University Hospital, Hiroshima, Japan, 2 Department of Endoscopy, Hiroshima University Hospital, Hiroshima, Japan, 3 Department of Gastroenterology and Metabolism, Graduate School of Biomedical and Health Sciences, Hiroshima University, Hiroshima, Japan, 4 Department of Health and Science, Prefectural University of Hiroshima, Hiroshima, Japan

* r-hayashi@hiroshima-u.ac.jp

**Data Availability Statement:** All relevant data are within the manuscript and its Supporting Information files.

## Abstract

Serum amyloid A (SAA) is an acute phase inflammatory protein that we previously described as a robust biomarker of colorectal inflammation in patients with ulcerative colitis (UC) in clinical remission. However, what induces SAA expression in UC remains unclear. This study demonstrates that SAA is significantly expressed in the intestinal tract of UC mouse models when compared with C-reactive protein, another inflammatory biomarker. Moreover, interleukin-6 and tumor necrosis factor-α were found to promote SAA1 expression, as were Toll-like receptor ligands flagellin and lipopolysaccharide. Furthermore, results suggested that the nuclear factor-kappa B (NF-κB) pathway may be involved in the promotion of SAA1 expression by flagellin, which was inhibited by treatment with 5-aminosalicylic acid (5-ASA). Therefore, the flagellin/NF-κB/SAA1 axis may represent one of the mechanisms by which 5-ASA suppresses intestinal inflammation.

## Introduction

Serum amyloid A (SAA) is a 12 kDa acute phase protein encoded by the *SAA* genes located on human chromosome 11, among which *SAA1* and *SAA2* encode acute phase proteins, *SAA3* is a pseudogene, and *SAA4* is constitutively expressed and is not definitively associated with inflammation [1]. SAA expression is usually induced in response to infection or acute injury and promotes inflammation through the induction of inflammatory cytokine production and subsequent recruitment of granulocytes, monocytes, and T lymphocytes [2]. It is believed that SAA promotes T-helper 17 (Th17) differentiation and induces inflammation [3, 4]. Th17 cells and other interleukin (IL)-17-producing T cells play important roles at the intestinal mucosal surface and contribute to the regulation of symbiotic organisms that constitute the microflora by protecting them from pathogenic bacteria and fungi [5]. However, excessive Th17 responses can promote autoinflammatory diseases such as Crohn's disease, rheumatoid arthritis, and multiple sclerosis [6]. We previously reported that SAA performs well as a biomarker

**Funding:** The authors received no specific funding for this work.

**Competing interests:** The authors have declared that no competing interests exist.

of endoscopic mucosal activity in clinical remissive (Rachmileswitz clinical activity index of 4 or less) ulcerative colitis (UC) [7]. Endoscopic inflammation is often experienced even in during clinical remission of UC. Such patients may not exhibit elevated C-reactive protein (CRP), rendering CRP ineffective as a biomarker but SAA can compensate for this issue. In addition, it has been reported that SAA is a better inflammation biomarker than C-reactive protein (CRP) in diseases other than UC [8]. The mechanism underlying SAA expression in the liver has only been investigated in hepatocellular carcinoma cell lines [9]. Thus, whether the same mechanism acts in normal cells remains unclear. SAA expression in the intestinal epithelium has been confirmed by immunostaining and *in situ* hybridization [4, 10]. However, knowledge gaps remain regarding the mechanism of SAA expression in the intestinal epithelium. SAA expression was promoted by IL-22 in normal mouse cells [11], by lipopolysaccharide (LPS) in mouse colorectal cancer cell lines [10], as well as by a combination of IL-1ß, IL-6, and tumor necrosis factor-α (TNF-α) in human colorectal cancer cell lines [12]. The purpose of this study was to determine the main organ of SAA expression in enteritis as well as the underlying mechanism of SAA expression in the normal intestinal tract. To shed light into this matter, we compared the expression of SAA and CRP using a dextran sulfate sodium (DSS)-induced enterocolitis mouse model, and investigated whether SAA expression could be promoted in normal cells by stimulating small intestinal organoids with various cytokines and Toll-like receptor (TLR) ligands, such as flagellin and LPS. Human SAA3 is a pseudogene, whereas mouse SAA3 is expressed by adipocytes and macrophages. However, human and mouse SAA1 are highly homologous and are often used as research models [13]. In addition, SAA1 works mainly as an inflammatory protein. This study primarily focuses on the role of SAA1 in enteritis, and the expression levels of the subtypes SAA2-4 were also examined.

## Materials and methods

### Antibodies and reagents

Recombinant murine IL-1β (#211-11B), IL-6 (#216–16), IL-10 (#210–10), and IL-22 (#210–22) were purchased from PeproTech (Cranbury, NJ, USA). Recombinant mouse IL-23 (#1877-ML) and anti-SAA1 goat antibody (#AF2948) were purchased from R&D Systems (Minneapolis, MN, USA). Flagellin from *Salmonella typhimurium* (#tlrl-stfla) was purchased from InvivoGen (San Diego, CA, USA). LPS was purchased from Sigma-Aldrich (St. Louis, MO, USA). TNF-α (#203–14261) was purchased from Fujifilm Wako Pure Chemical Corporation (Osaka, Japan). Anti-p-NF-κB rabbit antibody (#3033) and inhibitor kappa Bα (IκBα) antibody (#9242) were purchased from Cell Signaling Technology (Danvers, MA, USA). 5-aminosalicylic acid (5-ASA) was provided by Kyorin Pharmaceutical Co. (Tokyo, Japan), and was dissolved in culture medium at 40 mmol/L (pH 7.2 adjusted with NaOH) as described previously [14]. The NF-κB inhibitor BAY11-7082 (#T2846) was purchased from Tokyo Kasei (Tokyo, Japan).

### Mice

Specific pathogen-free C57BL/6 (B6) mice were purchased from CLEA Japan (Tokyo, Japan). All mice were housed under pathogen-free conditions in microisolator cages in the animal facility at the Hiroshima University, under 12 h light-dark cycles with access to water and food *ad libitum*. The health of the mice was monitored every day, and the animals were maintained in accordance with the Guidelines for the Care and Use of Laboratory Animals established by the Hiroshima University. This study was approved by the Committee on the Ethics of Animal Experiments of Hiroshima University (Permit Number: A18-27). All mice underwent cervical

dislocation euthanasia after administration of medetomidine hydrochloride, midazolam, and butorphanol.

### DSS-induced colitis model

Female mice (7 weeks of age) were divided into two groups (6 mice/group) that were treated with or without 2.5% DSS (MW: 5 kDa; Wako Chemical, Osaka, Japan) in the drinking water for 7 days. After completion of the treatment, the animals were sacrificed, and the liver, terminal ileum, ascending colon, and rectum were collected.

Two replicates of the DSS-induced colitis model experiment were performed and results were found to be reproducible.

### Protein analysis

Total protein was extracted from the small intestine organoid using RIPA lysis and extraction buffer (Thermo Fisher Scientific Waltham, MA #89901) with protease inhibitor (Roche, Basel, Switzerland #10276200). After incubation for 5 min at room temperature, cell lysates were centrifuged at 12,000 rpm and 4˚C for 20 min to collect protein lysates. After determination of protein concentration using a Pierce BCA Protein Assay Kit (Thermo Fisher Scientific #89901), automated quantitative western blotting (Wes assay) was performed on a Wes instrument (Protein Simple, San Jose, CA) according to the manufacturer instructions. GAPDH, phosphorylated-p65, and IκBα antibodies were used at a dilution of 1:100.

### Real-time polymerase chain reaction (PCR)

RNA was isolated from different organ and organoid samples using a RNeasy Mini kit (Qiagen, Hilden, Germany) according to the manufacturer's instructions. cDNA was synthesized from 1,000 ng of total RNA using a Reverse Transcription Kit (Qiagen), and quantitative PCR was performed with the SYBR Green Master-Mix (Qiagen) using a Light Cycler (Roche, Basel, Switzerland) according to the manufacturer's recommended protocol. Samples were normalized for *ACTB* expression in each organ and small intestinal organoid. The primer sequences used were as follows: *SAA1* forward 5′–AAATCAGTGATGGAAGAGAGGC–′ 3 and reverse 5′–CCCCAGCACAACCTACTGAG–′ 3; *SAA2* forward
5′–TGCTGAGAAAATCAGTGATGCAA–′ 3 and reverse
5′–CCCAACACAGCCTTCTGAAC–′ 3; *SAA3* forward
5′–AAGAAGCTGGTCAAGGGTCTA–′ 3 and reverse
5′–TCTTTTAGGCAGGCCAGCAG–′ 3; *SAA4* forward
5′–GGGAGGTCTTGCTCGTGATT–′ 3 and reverse
5′–AAGTCCCAAGTCCCTTGTACG–′ 3; *SAA4* forward
5′–GGGAAGCCGTACAAGGGACT–′ 3 and reverse
5′–CCTCGGGTCGGAAGTGATTG–′ 3; *CRP* forward 5′–CAGATCCCAGCAGCATCCA T–′ 3 and reverse 5′–CCTTTTTAAACATGTCTTCATGACC–′ 3; *ACTB*/S 5′–AGATCAAG ATCATTGCTCCTCCT–′ 3 and *ACTB*/SA 5′–ACGCAGCTCAGTAACAGTCC–′ 3.

### Immunofluorescence staining

Frozen specimens cut into 4-μm sections were placed on glass slides and fixed for 15 min in 4% paraformaldehyde in phosphate-buffered saline (PBS) solution. The slides were briefly blocked in protein blocking solution and incubated with an anti-SAA1 goat antibody (1:50 dilution) and anti-p-NF-κB rabbit antibody (1:200 dilution) overnight at 4˚C. The slides were washed with PBS and incubated for 1 h at room temperature with Alexa Fluor 488- and

568-labeled secondary antibodies. Nuclei were counterstained with 4',6-diamidino-2-pheny-lindole for 10 min, and mounting medium was placed on each specimen with a glass coverslip. SAA1 was identified by red fluorescence, whereas p-NF-κB was identified by green fluorescence. Negative controls were photographed under the same conditions (exposure time, etc.) to confirm the absence of autofluorescence.

## Murine organoid culture establishment and maintenance

Small intestine tissue collected from euthanized mice was used to establish organoids referring to the method reported by Sato et al. [15]. The fat and blood vessels were removed. The distal jejunum was collected (approximately 5 cm), and the lumen of the intestine was washed with PBS to remove contents and mucus. Next, the tissue was finely chopped, thoroughly washed with ice-cold PBS, placed in a 2.5 mM EDTA/PBS solution, and incubated in a refrigerator at 4°C for 30 min. Intestinal crypts were sequentially extracted by five rounds of mechanical shearing in PBS. After filtration using a 70 μm filter, the cells were washed with PBS and pelleted (300 μg, 3 min, 4°C). The isolated intestinal crypt was embedded in a 35 μL Matrigel (Corning, Bedford, MA, USA) dome and stored in complete IntestiCult organoid growth medium (Stemcell Technologies, Vancouver, Canada) at 37°C and 5% $CO_2$. Immediately before use, the desired antibiotics were added to fresh or thawed IntestiCult Organoid Growth Medium supplemented with 50 μg/mL gentamicin (nacalai tesque, Kyoto, Japan) and 100 units/100 μg per mL penicillin/streptomycin (Wako Chemical). The medium was replaced every 2–3 days, and the organoids were subcultured every 7–9 days by mechanical shearing and reimplantation in the 35 μL Matrigel dome at a split ratio of 1:2 to 1:3.

## Cell stimulation

To investigate the effect of cytokines on *SAA* expression, small intestinal organoids were stimulated with IL-1β (100 ng/mL), IL-6 (10 ng/mL), IL-10 (50 ng/mL), IL-22 (50 ng/mL), IL-23 (50 ng/mL), or TNF-α (10 ng/mL) for 24 h. In addition, the organoids were exposed to flagellin (10 μg/mL) or LPS (1 μg/mL) for 3 h, respectively, to investigate the effect of TLR ligands on the expression of *SAA*. The BAY11-7082 (20 μM) inhibitor was used to assess involvement of the NF-κB pathway in *SAA* expression, and 5-ASA (40 mM), commonly used for UC treatment, was also administered. The intestinal organoids were incubated with each inhibitor 1 h prior to adding flagellin.

## Statistical analyses

All statistical analyses were performed using EZR (Saitama Medical Center, Jichi Medical Center), a graphical user interface for R (The R Foundation for Statistical Computing, version 2.13.0) [16]. The Mann–Whitney *U* test was performed to compare multiple groups. A P-value $< 0.05$ was considered to indicate a statistically significant difference.

# Results

## DSS-induced colitis enhances intestinal *SAA* expression *in vivo*

Using the DSS enterocolitis mouse model, the expression of *SAA1-4* and *CRP* was evaluated in the liver and several parts of the intestinal tract to determine the main organ of SAA expression in enteritis. We hypothesized that the high sensitivity of SAA was a result of SAA being expressed primarily in the intestinal tract; however, SAA1-4 was expressed primarily in the liver (Fig 1A). Next, the expression of *SAA1-4* in the liver, terminal ileum, ascending colon, and rectum was compared between the control group (untreated mice) and DDS-treated

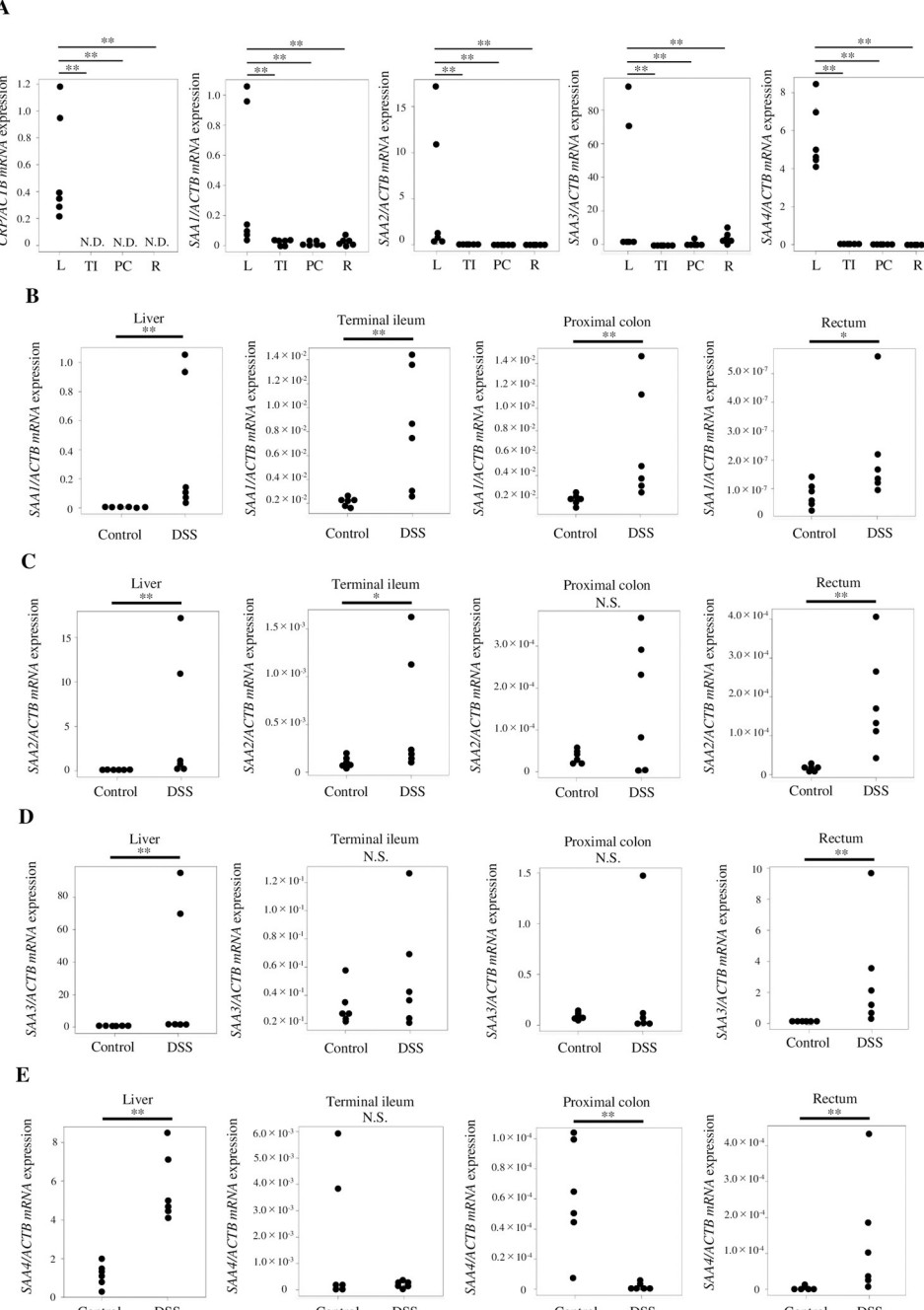

**Fig 1. Expression of *SAA1-4/CRP* in the liver and intestine of mice with dextran sulfate sodium (DSS)-induced colitis.** (A) Normalized *CRP* and *SAA* expression levels in the liver and each part of the intestinal tract 7 days after DSS administration. (B) After DSS administration, *SAA1* expression level increased in the liver, terminal ileum, ascending colon, and rectum when compared to untreated controls. (C) *SAA2* was significantly upregulated in organs other than the ascending colon. (D) *SAA3* expression level was significantly increased in the liver and rectum but not in the terminal ileum and ascending colon. In particular, *SAA3* expression level was strongly increased in the rectum. (E) *SAA4* expression was significantly upregulated in the liver and rectum. On the contrary, in the ascending colon, treatment with DSS suppressed the expression of *SAA4*. Data are represented as the mean ± standard deviation (n = 6) upon normalization to *ACTB* expression. **P < 0.01 and *P < 0.05 versus untreated controls. N.D., not detected; L, liver; TI, terminal ileum; PC, proximal colon; R, rectum.

model mice. *SAA1* expression level was higher in the DSS group for all tested samples (Fig 1B), suggesting that DSS-induced inflammation in the intestinal tract significantly enhanced local *SAA1* expression. *SAA2* was significantly upregulated in organs other than the ascending colon (Fig 1C). *SAA3* expression level was not increased in the terminal ileum and ascending colon but was significantly increased in the liver and rectum. In particular, *SAA3* expression level was strongly increased in the rectum (Fig 1D). *SAA4* expression was also significantly upregulated in the liver and rectum (Fig 1E). In contrast, DSS treatment suppressed *SAA4* expression in the ascending colon (Fig 1E). However, it is not surprising that the expression level of *SAA4* was reduced because *SAA4* expression is not associated with inflammation.

## Inflammatory cytokines promote *SAA* expression in small intestinal organoids

Next, we evaluated whether different cytokines could impact the expression of *SAA* in mouse small intestinal organoids. In addition to IL-22, which upregulates SAA1 as previously reported [11], our study revealed that IL-6 and TNFα upregulate *SAA1* in the intestinal epithelium (Fig 2A, 2B, and 2E). No statistically significant difference was observed for IL-1β, however, *SAA1* expression level tended to increase in the presence of this cytokine (Fig 2C). *SAA1* expression level remained unaltered in the presence of IL-23 (Fig 2D). IL-1β (1, 50, 100 ng/mL) and IL-23 (10, 50, 100 ng/mL) levels were also tested and similarly showed no significant induction of SAA in intestinal organoid cultures. In contrast, *SAA1* expression was inhibited by IL-10, but this result was not statistically significant (Fig 2F), which is consistent with previous reports demonstrating that IL-10 is an anti-inflammatory cytokine that can suppress colitis [17, 18]. It was difficult to evaluate *SAA2* and *SAA4* expression because their expression levels were not quantifiable (Fig 2A–2F). The expression level of *SAA3* was significantly increased by IL-22 and IL-23 (Fig 2D and 2E). However, unexpectedly, *SAA3* was downregulated by IL-1β (Fig 2C).

## Toll-like receptor (TLR) stimulation promotes *SAA* expression *in vitro* via the NF-κB pathway

TLR4, which recognizes LPS (a component of the outer membrane of Gram-negative bacteria), and TLR5, which recognizes flagellin (a major structural protein in bacterial flagella), have been reported to induce inflammatory cytokine production via NF-κB [19–23]. NF-κB was also previously reported to be involved in the induction of SAA expression [24]. Therefore, we questioned whether SAA expression in intestinal epithelial cells could be enhanced via TLR-stimulated NF-κB signaling triggered by flagellin or LPS. The addition of flagellin or LPS to small intestinal organoid cultures significantly increased the expression level of *SAA1/3* (Figs 3A and S1A). Interestingly, the presence of an NF-κB inhibitor (BAY11-7082) suppressed the flagellin-induced *SAA1/3* expression (Fig 3A). Immunofluorescence staining also showed that flagellin strongly induced SAA1 expression, and NF-κB inhibitors suppressed flagellin-induced SAA expression in intestinal epithelial cells (Fig 3B). As previously reported, SAA3 was induced by flagellin via NF-κB, and the same pathway was thought to be involved in SAA1 induction [25]. In contrast to the flagellin pathway, LPS-mediated SAA expression is NF-κB-independent (S1B Fig). The expression of SAA2 and SAA4 was not observed after stimulation with LPS or flagellin (Figs 3A and S1A).

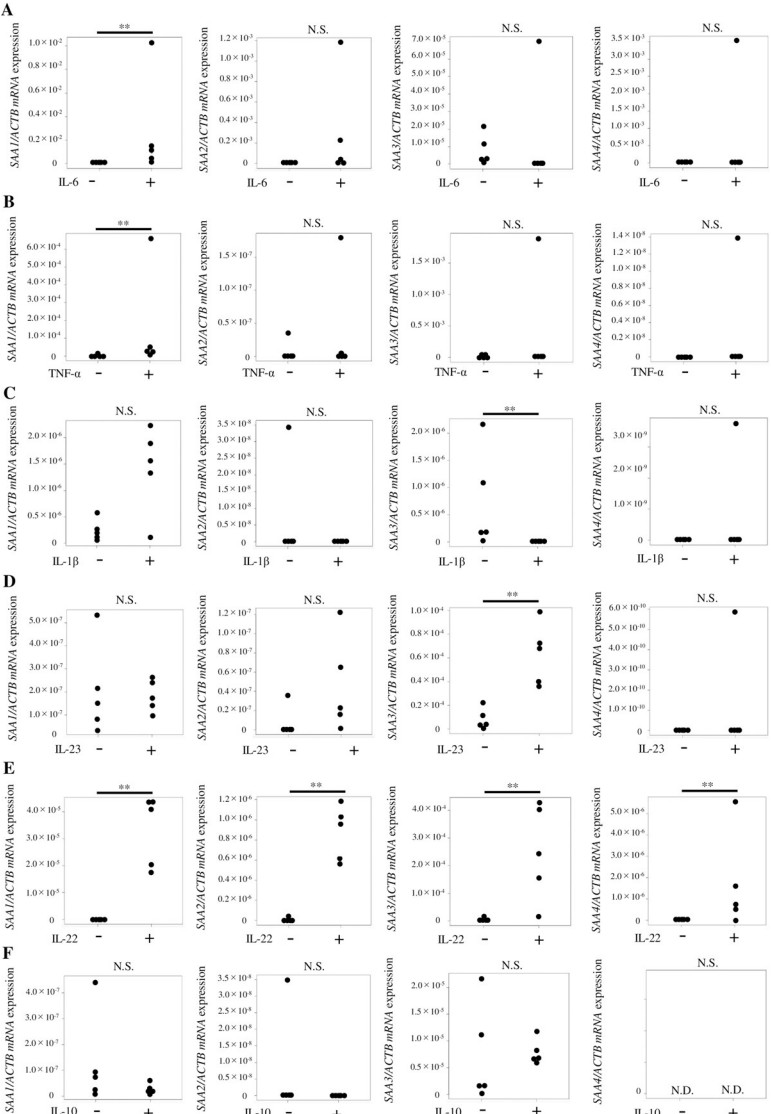

**Fig 2. *SAA1-4* expression in mouse small intestinal organoids upon stimulation with several cytokines.**
Normalized *SAA* expression levels in small intestinal organoids upon stimulation with (A) IL-6 (10 ng/mL), (B) TNF-α
(10 ng/mL), (C) IL-1β (100 ng/mL), (D) IL-23 (50 ng/mL), (E) IL-22 (50 ng/mL), or (F) IL-10 (50 ng/mL) for 24 h.
Only IL-6, TNF-α, and IL-22 promoted a significant increase in *SAA1* expression level when compared with untreated
controls. Although not statistically significant, the expression level of *SAA1* tended to increase upon IL-1β stimulation
and was suppressed by stimulation with IL-10. It was difficult to evaluate *SAA2* and *SAA4* expression because their
expression levels were not detectable (A-F). The expression level of *SAA3* was significantly increased by IL-22 and IL-
23 (D, E). However, unexpectedly, *SAA3* was downregulated by IL-1β (C). *SAA* expression level was normalized to that
of *ACTB* and is represented as the mean ± standard deviation (n = 5). **P < 0.01 versus untreated controls. N.S., not
significant; N.D., not detected.

## 5-ASA suppresses the production of *SAA1* via NF-κB

In previous reports, NF-κB was suppressed by 20–40 mM 5-ASA in HCT-116 and Caco-2
human colon cancer cell lines [26, 27]. Based on these findings, SAA expression was evaluated
in the presence of flagellin with or without 40 mM 5-ASA. mRNA analysis revealed that *SAA1/
3* expression was suppressed by the presence of 5-ASA (Fig 4A). In normal cells, NF-κB is
mainly found in the cytoplasm in its inactive form, associated with inhibitory proteins I kappa

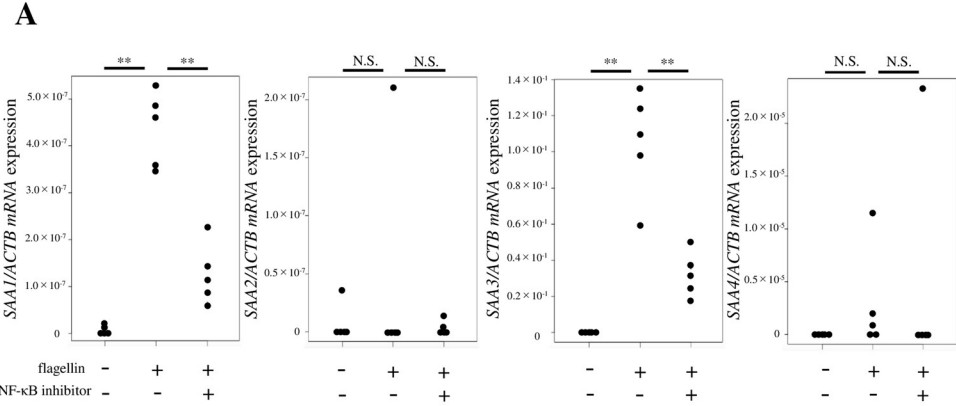

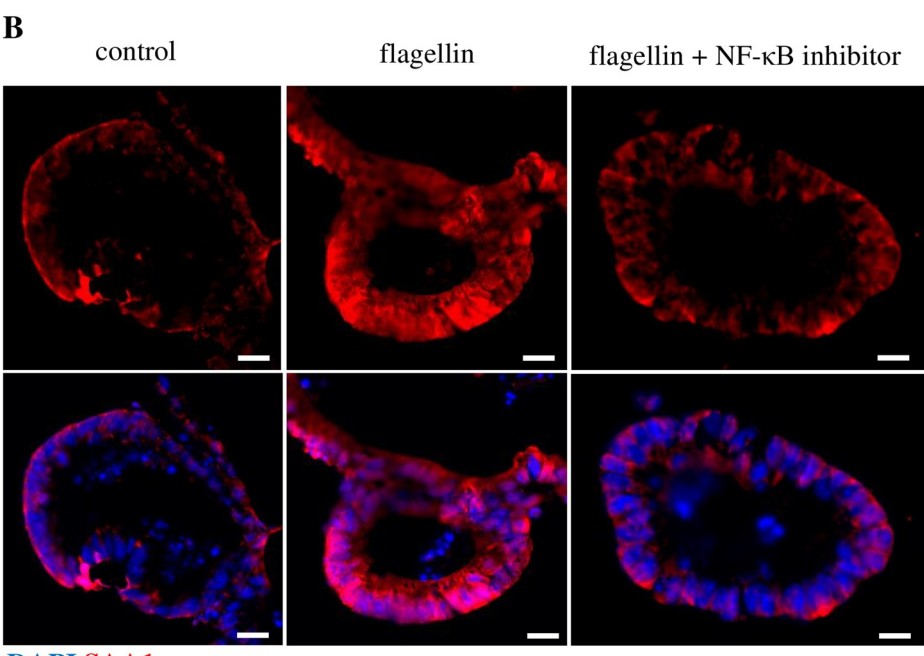

DAPI SAA1

**Fig 3. Flagellin promotes *SAA1-4* expression in small intestinal organoids via NF-κB.** (A) Normalized *SAA* expression levels in small intestinal organoids upon stimulation with Toll-like receptor ligands flagellin (10 μg/mL) for 3 h. The addition of flagellin significantly increased the expression level of *SAA1/3*, an effect that was inhibited by NF-κB inhibitor BAY11-7082 (20 μM). *SAA* expression level was normalized to that of *ACTB* and is represented as the mean ± standard deviation (n = 5). **P < 0.01. (B) Immunostaining of mouse small intestinal organoids confirmed that flagellin enhanced SAA1 levels. This effect was suppressed in the presence an NF-κB inhibitor. All samples were photographed under the same conditions. Negative controls were also photographed under the same conditions (exposure time, etc.) to confirm the absence of autofluorescence. Scale bar, 10 μm.

B (IκB)-α and p100. However, upon receiving the activation signal, inhibitory proteins are degraded, while NF-κB is activated and translocates into the nucleus where it functions as a transcription factor [28]. In fact, the IκB protein degradation was observed after 30 min of flagellin stimulation of small intestinal organoids (Fig 4B). The protein expression level of IκB was then increased at 3 h after flagellin stimulation, which may be because IκB is restored at 2 h, as previously reported [27]. In addition, sulfasalazine reportedly represses IκBα gene expression [29], and the protein expression of IκBα was also repressed by 5-ASA in the present study

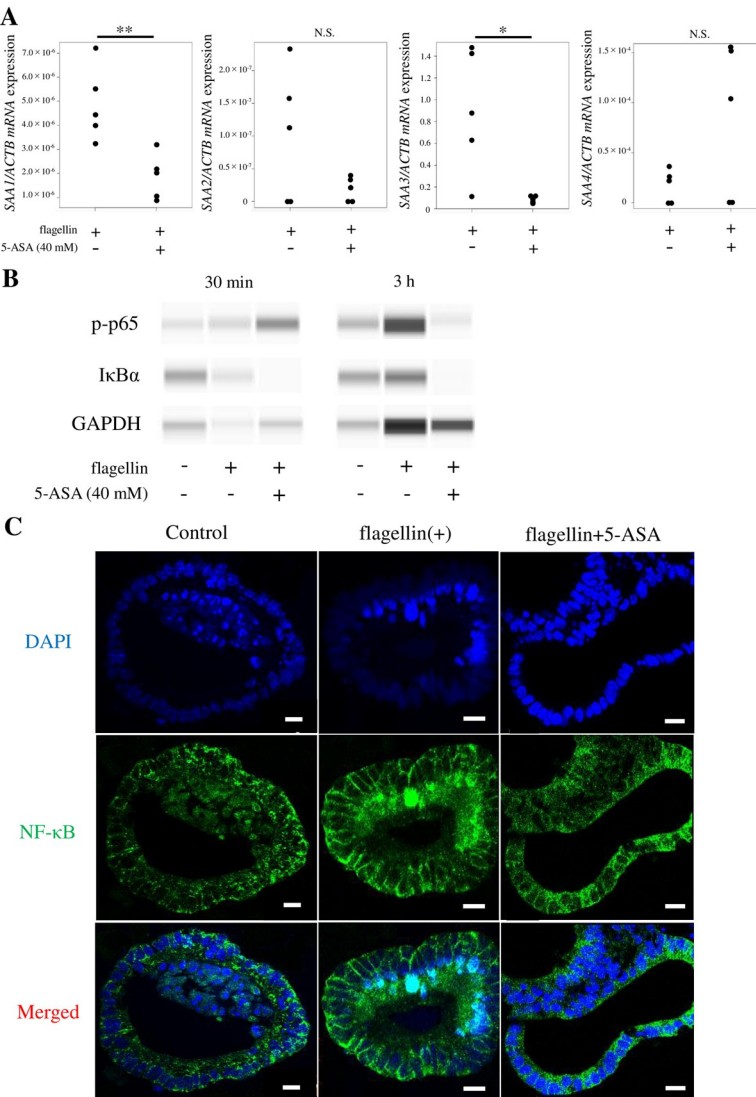

**Fig 4. 5-Aminosalicylic acid (5-ASA) suppresses *SAA1-4* expression via NF-κB in small intestinal organoids.** (**A**) Normalized *SAA* expression levels in small intestinal organoids upon stimulation with flagellin alone (10 μg/mL) or plus 5-ASA (40 mM) after 3 h. *SAA1/3* was suppressed by the addition of 5-ASA. *SAA* expression level was normalized to that of *ACTB* and is represented as the mean ± standard deviation (n = 5). **P < 0.01, *P < 0.05. (**B**) Small intestinal organoids were stimulated with flagellin at 30 min and 3 h with and without 5-ASA. The protein expression levels of phosphorylated-p65 and IκB were examined by SimpleWestern™. The protein degradation of IκB was observed after 30 min of flagellin stimulation of small intestinal organoids, and the protein expression of IκBα was also repressed by 5-ASA. The inhibition of p65 phosphorylation by 5-ASA was not apparent at 30 min but clear 3 h after flagellin stimulation. This experiment was performed twice independently, and similar results were obtained. A scan of the original untrimmed gel image is displayed in S2 Fig. (**C**) Flagellin promotes activation of the NF-κB pathway, as observed by immunostaining data showing translocation of NF-κB into the nucleus. Immunofluorescence confirmed that NF-κB translocation into the nucleus was inhibited with the addition of 5-ASA. Each negative control was photographed under the same conditions (exposure time, etc.) to confirm the absence of autofluorescence. Scale bar: 10 μm.

(Fig 4B). It has been reported that 5-ASA exerts its anti-inflammatory effect by inhibiting NFκB phosphorylation [27]. In this study, the inhibition of p65 phosphorylation by 5-ASA was not apparent at 30 min but clear 3 h after flagellin stimulation. (Fig 4B). When TLR stimulation was induced via flagellin in the small intestinal organoid, immunofluorescence revealed

that NF-κB translocated into the nucleus (Fig 4C). Taken together, these results suggested that flagellin-induced SAA expression in normal small intestinal epithelium is mediated by the NF-κB pathway. Furthermore, immunofluorescence revealed that NF-κB translocation into the nucleus was suppressed by the addition of 5-ASA (Fig 4C). Thus, 5-ASA prevented NF-κB nuclear translocation, in turn suppressing SAA expression.

## Discussion

To date, it has been reported that SAA is a superior inflammatory biomarker to CRP in certain diseases [8]. We previously reported that SAA performed well as a biomarker of endoscopic mucosal activity in clinical remissive UC [7]. This led us to question why SAA was more sensitive than CRP in clinical remissive UC. One possibility is that SAA expression level in the intestinal tract is higher than CRP expression level. To address this, the expression levels of SAA and CRP in the intestinal tract of DSS-induced enterocolitis mouse models were detected. Although SAA expression was mainly detected in the liver, it was also found to be significantly upregulated in the intestinal tract upon DSS treatment, in contrast to CRP which was not expressed in the intestinal tract. When UC is active, cytokines produced in the intestinal tract can reach the liver via the portal vein, potentially inducing the expression of CRP. However, in clinical remissive mild UC, the inflammatory process is localized to the intestinal tract, and thus, cytokines may remain there rather than travel throughout circulation. This may be one of the reasons for the superior performance of SAA as a biomarker of intestinal inflammation in clinical remissive UC.

The regulatory mechanisms controlling SAA expression are poorly known and may differ between cancer and normal cells. To unveil these molecular mechanisms, mouse small intestinal organoids composed of normal cells were used. Stimulation of organoids with cytokines, such as IL-6, IL-22, and TNF-α, as well as TLR ligands (flagellin and LPS) promoted *SAA1* expression, suggesting that SAA could be produced independently in the intestine. A previous study had already demonstrated that IL-22 could enhance SAA expression in small intestinal organoids [11]. However, the data described herein reveals the involvement of additional cytokines in the regulation of SAA production. Therefore, the signaling network underlying SAA regulation may be more complex than initially presumed. In addition, the promotion of SAA production by flagellin and LPS is thought to be involved in immune activation by bacteria. Segmented filamentous bacteria (SFB), which engraft in the intestinal epithelium, were described to induce SAA expression through the activation of innate lymphoid cells and the promotion of Th17 cell differentiation [11]. Moreover, there are also reports that SFB contains flagellin [30], which may be involved in SAA production.

Further experiments revealed that the expression of SAA was suppressed by adding 5-ASA to the small intestinal organoid. 5-ASA is known to inhibit the phosphorylation of NF-κB at certain concentrations [26, 27], in turn downregulating SAA production.

The effect of 5-ASA on nuclear translocation of NF-κB is currently unknown due to contradictory experimental findings. A previous report has stated that 5-ASA does not inhibit nuclear translocation [29], whereas another paper demonstrated nuclear translocation by 5-ASA in a cancer cell line [31]. In this paper, we demonstrate that 5-ASA inhibits the nuclear translocation of p65 in a normal cell line. We also hypothesize that *SAA* is a direct target gene of NFκB, as it has been previously shown that the NFκB binding region (5'-GGGACTTTCC-'3) is present in the enhancer region of the *SAA* gene [32]. 5-ASA is the most used drug for the treatment of UC, suppressing inflammation of the intestinal tract. The data described herein revealed that one of the mechanisms by which 5-ASA exerts its anti-inflammatory effects may

be through the direct suppression of SAA expression in the intestinal epithelium, similarly to anti-inflammatory cytokine IL-10.

In conclusion, in a mouse model of DSS enteritis, SAA is also expressed in the intestinal tract but is mainly expressed in the liver. Proinflammatory cytokines such as IL-6, TNF-alpha, and IL-22 induce SAA in normal intestinal epithelium. TLR stimulation with flagellin induces SAA via NF-κB, and 5-ASA inhibits this pathway and suppresses SAA expression. The non-NF-κB-dependent pathway that may be involved in the induction of SAA expression by LPS is a subject of future research.

## Supporting information

**S1 Fig. The lipopolysaccharide (LPS)-induced increase in serum amyloid A (SAA) expression is independent of the NF-κB pathway.** The addition of LPS significantly increases the expression level of *SAA1/3* (A), an effect that is not inhibited by the NF-κB inhibitor, BAY11-7082 (B). The *SAA* expression level was normalized to that of *ß-actin (ACTB)* and is presented as the mean ± standard deviation (n = 5). **P < 0.01.
(TIF)

**S2 Fig. Scan of the original untrimmed gel image of Fig 4B.**
(TIF)

## Acknowledgments

We would like to thank Editage (www.editage.jp) for English language editing.

## Author Contributions

**Conceptualization:** Masaki Wakai, Ryohei Hayashi, Yasuhiko Kitadai, Shinji Tanaka.

**Data curation:** Masaki Wakai, Kana Onishi, Takeshi Takasago, Hidehiko Takigawa.

**Formal analysis:** Masaki Wakai, Ryohei Hayashi, Yoshitaka Ueno, Takuro Uchida, Yuji Urabe, Shiro Oka.

**Investigation:** Takeshi Takasago, Takuro Uchida, Hidehiko Takigawa, Ryo Yuge, Yuji Urabe.

**Methodology:** Yoshitaka Ueno, Ryo Yuge.

**Supervision:** Ryohei Hayashi, Shiro Oka, Shinji Tanaka.

**Validation:** Takuro Uchida.

**Writing – original draft:** Masaki Wakai.

**Writing – review & editing:** Masaki Wakai, Kana Onishi.

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
