## [Decision Letter · Decision Letter 0]

26 Aug 2021

PONE-D-21-23414

Promoting mechanism of serum amyloid A family expression in mouse intestinal epithelial cells

PLOS ONE

Dear Dr. Hayashi,

Thank you for submitting your manuscript to PLOS ONE. After careful consideration, we feel that it has merit but does not fully meet PLOS ONE’s publication criteria as it currently stands. Therefore, we invite you to submit a revised version of the manuscript that addresses the points raised during the review process.

We look forward to receiving your revised manuscript.

Kind regards,

Wendy Huang, Ph.D.

Academic Editor

PLOS ONE

Journal Requirements:

2. Thank you for including your ethics statement in the online submission form. Please ensure you include this statement (both the name of the ethics committee that approved your study, and the method of euthanasia) in the manuscript Methods.

Reviewers' comments:

Reviewer's Responses to Questions

**Comments to the Author**

1. Is the manuscript technically sound, and do the data support the conclusions?

Reviewer #1: Partly

Reviewer #2: Partly

2. Has the statistical analysis been performed appropriately and rigorously? 

Reviewer #1: No

Reviewer #2: No

3. Have the authors made all data underlying the findings in their manuscript fully available?

Reviewer #1: Yes

Reviewer #2: Yes

4. Is the manuscript presented in an intelligible fashion and written in standard English?

Reviewer #1: No

Reviewer #2: Yes

5. Review Comments to the Author

Reviewer #1: The authors previously published their finding that SAA can be used better for biomarker as well as endoscopic mucosal activity in ulcerative colitis rather than well-used biomarkers, such as CRP. In this manuscript, they aimed to determine the major organs of SAA expression as well as the underlying mechanisms of SAA expression in the GI tract using the DSS mouse model. To improve it, there were several concerns in this manuscript.

Major

I would like to know that the experiment was repeated, especially DSS colitis. If it was repeated, please mention that in the text (Materials and Methods). The error bars of SD was relatively big. Showing each value as a dot must be the best approach for all the readers. Please show all the RT-PCR data in the graphs using dot plots.

Please explain how the authors determined the concentration of these cytokines for the detection of SAAs. Basically, the authors should have had several titrations (dilution) to optimize the experiment. The concentration which showed saturated expression of SAA must not be used.

In this study, the best option was to use human organoids to check the SAAs expression; even the authors used the DSS animal model in Figure 1. Please consider doing this experiment. So far, only using cell line and mouse organoid are very weak.

The biggest concern (issue) in this paper was the analyzing method for the RT-PCR data. What does “normalized SAA expression” mean? All the data looks like a relative expression somewhere. Each data sets have 1.0 in somewhere. Did the author set one data set 1.0, and then relative levels were presented? In the method section, it was mentioned that samples were normalized for (must be by) ACTB expression in each organ and organoid). This was the right way. However, the expression of SAAs was quite low. For example, if you have just 0.00001 for SAA1 in untreated, then they should not use the relative expression. If the quite low value was set as 1.0, then the obtaining relative expression value must be really high, and all the samples are messed up and cannot be compared. Based on this point, all the data did not completely make sense. The authors must learn this and re-analyze all the RT-PCR data correctly.

Minor:

It would be interesting if the author shows the expression levels of cytokine receptors using organoids (ideally both mouse and human organoids).

Please show fluorescent minus one for the immunohistochemistry data for Figures 3C and 4B)

Please provide the information about 5-ASA. Need to explain in more detail why the authors used this drug? What are the differences between NFkb inhibitors (BAY11-7082) and 5-ASA? The author mentioned that In previous reports, NF-κB was suppressed by 20–40 mM 5-ASA in HCT-116 and Caco-2 human colon cancer cell lines. Then what was the purpose of the use of 5-ASA? It sounds just like a very similar experiment.

Reviewer #2: In this manuscript, Wakai et al. present data which leads them to conclude that the expression of serum amyloid A (SAA), in particular SAA1, is regulated by the NF-kB signaling pathway through toll-like receptor (TLR) stimulation in ulcerative colitis (UC) mouse model. SAA is significantly expressed in the intestinal track of UC mouse model and a good inflammatory biomarker. However, the mechanism underlying SAA expression remains to be. Using a dextran sulfate sodium (DSS)-induced enterocolitis mouse model, the authors study SAA expression by various cytokines and TLR ligands. The pre-treatment with an NF-kB inhibitor, BAY11-7082, dampened the expression of SAA mRNA leads the authors to conclude that TLR stimulation induces SAA via NF-kB. This is an interesting topic; however, I found some of the data are relatively weak to support the conclusion with the following major concerns.

1) In Figure 3, the authors showed both LPS (through TLR4) and flagellin (through TLR 5) induced SAA1 and 3’s mRNA expression. However, the NF-kB inhibitor BAY11-7082 treatment was only done in flagellin (Fig 3B) but LPS (Fig 3A) stimulation. LPS is known to activate both canonical and non-canonical NF-kB signaling. Do the authors have any idea/data suggesting with pathways it is?

2) After pre-treatment with BAY11-7082, the NF-kB activities was not monitored. BAY11-7082 was known to inhibit IkBalpha phosphorylation, as well as directly inhibit functions on the NLRP3 inflammasome by blocking the sensor's ATPase activity. Therefore, showing the IkBa’s protein degradation and NF-kB nuclear translocation is necessary to support the inhibitory effects of SAA expression was through NF-kB signaling.

3) If the expression of SAA is regulated by NF-kB, is SAA a direct target gene of NF-kB or it is a secondary effect of NF-kB activation? If it is a direct target gene, what is the kB binding site(s)?

4) The authors only showed SAA mRNA expression level upon different cytokines and TLR ligands stimulation, how about the SAA proteins’ expression level? It will be interest to see the protein level as well.

5) In general, all the gene expression figures have huge error, especially Figure 1 and 2; some panels the error bar is larger than the bar graph itself. Figure 1A does not have p-value either. Box & Whisker plots with overlay of individual data points are needed to show the significance of the data.

6. PLOS authors have the option to publish the peer review history of their article (what does this mean?). If published, this will include your full peer review and any attached files.

Reviewer #1: No

Reviewer #2: No

---

## [Author Response · Author response to Decision Letter 0]

26 Nov 2021

We have revised the manuscript in accordance with the suggestions made by the editor and reviewers. Please refer to the response to reviews for more details.

---

## [Editor Report · Decision Letter 1]

6 Jan 2022

PONE-D-21-23414R1Promoting mechanism of serum amyloid A family expression in mouse intestinal epithelial cellsPLOS ONE

Dear Dr. Hayashi,

Thank you for submitting your manuscript to PLOS ONE. After careful consideration, we feel that it has merit but does not fully meet PLOS ONE’s publication criteria as it currently stands. Therefore, we invite you to submit a revised version of the manuscript that addresses the points raised during the review process.

We look forward to receiving your revised manuscript.

Kind regards,

Wendy Huang, Ph.D.

Academic Editor

PLOS ONE

Journal Requirements:

Additional Editor Comments:

In the revised manuscript, Wakai et al completed additional experiments and revisions to address many of the reviewers’ main concerns. There are three remaining minor issues that the authors should address:

In response to Reviewer #1, point #2, the authors stated that “We examined the expression of SAA with these cytokines in a concentration-dependent manner… and found no significant difference”. No data figure were shown. The authors should at least include in the revised Result text section the exact range of the concentrations tested that yield the negative result. For example, “IL-1β (1, 100, 1000 ng/mL) and IL-23 (5, 50, 500 ng/mL) were also tested and similarly returned no significant induction of SAA in the intestinal organoid culture.”

Related to Reviewer #2, point #1: On page 12 of the text, the text stated “Figure 3. Flagellin and lipopolysaccharide (LPS) promote SAA1-4 expression in small intestinal organoid via NF-kB”. However, Fig 3 only included NF-kB inhibition results in the Flagellin condition, but not the LPS treatment. Therefore, Reviewer #2 asked NF-kB inhibitor to be tested on LPS treated cells. As the authors noted on page 59 of the point-by-point, result of this additional experiment suggest LPS induced SAA expression is NF-kB independent (data not shown). Given this new information, the authors should consider moving the Figure 3A(LPS) panel to a separate figure and take “LPS” out of the Figure 3 title/legend etc – the current writing would give readers the wrong impression that LPS induced SAA is dependent on NFkB. Instead, LPS+NFkB inhibitor results should be shown under a separate figure - texts in the results section should include description of the negative result (e.g. In contrast to the flagellin pathway, LPS induced SAA expression is NF-kB independent.). Discussion should be revised to comment on potential non-NFkB dependent pathway that maybe involved in LPS induced SAA expression subject to future studies.

Related to the author’s response to Reviewer #2, point #2: For Figure 4B, western experiment missing loading control (e.g. beta-actin or GAPDH). Figure legend needs to state how many time this experiment had been performed independently showing similar results. Original un-cropped gel images/scans should be displayed.
---

## [Author Response · Author response to Decision Letter 1]

2 Feb 2022

Reply:

We have confirmed that the reference list is complete and accurate.

In response to Reviewer #1, point #2, the authors stated that “We examined the expression of SAA with these cytokines in a concentration-dependent manner… and found no significant difference”. No data figure were shown. The authors should at least include in the revised Result text section the exact range of the concentrations tested that yield the negative result. For example, “IL-1β (1, 100, 1000 ng/mL) and IL-23 (5, 50, 500 ng/mL) were also tested and similarly returned no significant induction of SAA in the intestinal organoid culture.”

Reply:

Thank you very much for your valuable suggestion. We have added the following text to the Results section (Page 11, Lines 222–224):

“IL-1β (1, 50, 100 ng/mL) and IL-23 (10, 50, 100 ng/mL) levels were also tested and similarly showed no significant induction of SAA in intestinal organoid cultures.”

Related to Reviewer #2, point #1: On page 12 of the text, the text stated “Figure 3. Flagellin and lipopolysaccharide (LPS) promote SAA1-4 expression in small intestinal organoid via NF-kB”. However, Fig 3 only included NF-kB inhibition results in the Flagellin condition, but not the LPS treatment. Therefore, Reviewer #2 asked NF-kB inhibitor to be tested on LPS treated cells. As the authors noted on page 59 of the point-by-point, result of this additional experiment suggest LPS induced SAA expression is NF-kB independent (data not shown). Given this new information, the authors should consider moving the Figure 3A(LPS) panel to a separate figure and take “LPS” out of the Figure 3 title/legend etc – the current writing would give readers the wrong impression that LPS induced SAA is dependent on NFkB. Instead, LPS+NFkB inhibitor results should be shown under a separate figure - texts in the results section should include description of the negative result (e.g. In contrast to the flagellin pathway, LPS induced SAA expression is NF-kB independent.). Discussion should be revised to comment on potential non-NFkB dependent pathway that maybe involved in LPS induced SAA expression subject to future studies.

Reply:

We agree with the pertinent comments made by you. In accordance with your suggestion, the two graphs showing the change in SAA expression with LPS have been separated into separate figures (Supplement Fig 1A and 1B). We have also removed the word “LPS” from the title of Figure 3. In contrast to the flagellin pathway, the induction of SAA expression by LPS is NF-κB-independent; we have mentioned this in the Results section (Page 12, Lines 259–260). A mention of the need for further research to decipher the non-NF-κB-dependent pathway that may be involved in the induction of SAA expression by LPS has been made in the Discussion section (Page 17, Lines 367–368.)

Related to the author’s response to Reviewer #2, point #2: For Figure 4B, western experiment missing loading control (e.g. beta-actin or GAPDH). Figure legend needs to state how many time this experiment had been performed independently showing similar results. Original un-cropped gel images/scans should be displayed.

Reply:

Thank you for your valuable feedback. Figure 4B shows panels for GAPDH expression, used as a control. We have also mentioned in the text that this experiment was repeated twice and that the same trend was observed (Page 14, Line 309–310). The original, untrimmed gel image is shown in Supplement Figure 2, and the figure has been referred to in the manuscript (Page 14-15, Lines 310–311).

---

## [Editor Report · Decision Letter 2]

18 Feb 2022

Promoting mechanism of serum amyloid A family expression in mouse intestinal epithelial cells

PONE-D-21-23414R2

Dear Dr. Hayashi,

We’re pleased to inform you that your manuscript has been judged scientifically suitable for publication and will be formally accepted for publication once it meets all outstanding technical requirements.

Kind regards,

Wendy Huang, Ph.D.

Academic Editor

PLOS ONE
---

## [Editor Report · Acceptance letter]

10 Mar 2022

PONE-D-21-23414R2 

Promoting mechanism of serum amyloid A family expression in mouse intestinal epithelial cells 

Dear Dr. Hayashi:

I'm pleased to inform you that your manuscript has been deemed suitable for publication in PLOS ONE. Congratulations! Your manuscript is now with our production department. 

Kind regards, 

on behalf of

Dr. Wendy Huang 

Academic Editor

PLOS ONE